# Medication Adherence Barriers and Their Relationship to Health Determinants for Saudi Pediatric Dialysis Patients

**DOI:** 10.3390/children11030293

**Published:** 2024-02-29

**Authors:** Leena R. Baghdadi, Manar M. Alsaiady

**Affiliations:** 1Department of Family and Community Medicine, College of Medicine, King Saud University, Riyadh 11362, Saudi Arabia; 2King Khalid University Hospital, King Saud University Medical City, Riyadh 12372, Saudi Arabia; malsaiad@ksu.edu.sa

**Keywords:** barriers, medication adherence, pediatric, kidney failure, dialysis, determinants of health

## Abstract

Medication adherence is critical for the treatment and improved outcomes of chronic diseases. However, there is little research on the medication adherence of pediatric dialysis patients in Saudi Arabia. This study examines medication adherence barriers and their relationship to health determinants among Saudi children on dialysis, to enhance treatment success. We conducted a hospital-based, cross-sectional survey of pediatric dialysis patients using a simple random sampling technique. There is a trend of higher medication adherence for peritoneal dialysis patients compared with hemodialysis patients (36.1 ± 12.9 vs. 34.7 ± 8.3, *p* = 0.07). The leading barriers to medication adherence for all patients included being tired of taking the medication (score = 3.0256), not feeling like taking the medicine sometimes (score = 2.7436), bad taste (score = 2.5513), and forgetfulness (score = 2.41). Determinants of health were associated with medication adherence. Lack of education (56.4%) (some children underage for school) and chronic disease requirements (16.7%) were common barriers. After adjusting for the common confounders, the adherence scores increased significantly with increasing patient age (β = 2.378, *p* < 0.001), patients with working parents (β = 8.726, *p* = 0.011), and those living outside Riyadh (β = 19.198, *p* < 0.001). Medication adherence among pediatric dialysis patients is influenced by sociodemographic factors, health systems, and access to care. Evidence-based targeted interventions can increase medication adherence in this group on frequent dialysis. Future efforts should utilize systematic frameworks and digital health technologies to provide quality alternatives to improve medication adherence.

## 1. Introduction

Medication adherence is crucial for better medical outcomes, especially in chronic diseases, including chronic kidney disease (CKD). Medication adherence is the extent that patients follow treatment plans from medical professionals [1,2]. The World Health Organization defines it as the degree to which patients adhere to recommendations for medication, diet, and lifestyle changes [3]. Although used interchangeably with compliance, adherence implies collaboration between providers and patients, making it the preferred term [4]. Parameters to evaluate adherence in CKD and patients undergoing pediatric dialysis (hemodialysis [HD] and peritoneal dialysis [PD]) requiring complex lifestyle changes can be adapted from this definition.

CKD is an emerging public health issue in Saudi Arabia. The age-adjusted prevalence of CKD is approximated at 9892 per 100,000 individuals, surpassing the estimates for Western Europe (5446 per 100,000) and North America (7919 per 100,000) [1]. Dialysis plays a pivotal role in addressing kidney failure [1,5]. The country’s healthcare infrastructure has made significant strides in providing kidney replacement therapy, including HD and PD, to its population. However, there exists a compelling need to direct attention toward pediatric dialysis, an area that remains relatively underexplored. Limited research and clinical focus on pediatric dialysis in Saudi Arabia underscore the importance of comprehensive investigations into the unique challenges and clinical outcomes specific to children undergoing dialysis.

While dialysis is life-saving, and strict medical adherence is crucial, there are limited data on medication adherence barriers among pediatric dialysis patients and related health determinants [5]. Evidence shows lower adherence among vulnerable patients with chronic diseases compared to other conditions, as chronic illnesses are challenging due to their long-term presentation and multisystem consequences [6]. CKD significantly affects life expectancy, quality of life, and expenses, and factors differ in developing nations and children [6,7,8]. Identifying causes and mitigation strategies for medication adherence are public health priorities [9]. Early diagnosis and therapy are essential for pediatric CKD; however, monitoring medication adherence, patient counselling and education is often neglected in resource-limited settings, despite being linked to better outcomes [4].

Social, economic, and environmental determinants of health influence the healthcare access and health behaviors of patients with CKD [10]. Key barriers to medication adherence include demographics, socioeconomic status, adverse effects, and medical knowledge [11,12]. Research shows that non-adherence (non-intentional forgetting, lack of understanding, and intentional patient choice) stems from an inability to afford or access medicines [4].

Adherence to HD/PD is a major CKD treatment challenge. Studies show lower adherence rates by pediatric patients undergoing HD/PD in developing nations [13,14,15,16]. Among pediatric patients, non-adherence is 50–55% higher for chronic diseases compared to acute conditions [14]. However, research methods and non-adherence definitions vary [15]. Medical adherence has a major effect on outcomes but is complex for children with chronic diseases [16].

Research has consistently highlighted the challenge of low medical adherence among pediatric patients undergoing HD/PD [17]. The key determinants of medication adherence in this population encompass intricate factors such as family dynamics, physiological considerations, and coexisting illnesses [18]. In the context of developing nations, economic burdens, limited access to medications, and healthcare disparities compound the challenges faced by pediatric patients undergoing HD/PD, hindering favorable treatment outcomes [18]. Global studies further affirm the prevalence of suboptimal treatment adherence in pediatric kidney failure [19]. Notably, the existing literature highlights a critical gap in understanding the barriers to medication adherence, particularly in pediatric populations, with a predominant focus on adult patients [20]. This oversight limits the comprehension of unique challenges faced by pediatric patients undergoing HD/PD. Addressing this gap is imperative, as evidenced by the scarcity of public policy strategies aimed at enhancing access to complete treatment regimens, thereby alleviating economic strain on caregivers and potentially ameliorating adherence outcomes [21]. Recognizing and addressing these gaps in the literature will provide valuable insights to improve clinical outcomes for this vulnerable pediatric population, forming the aim for our hospital-based, cross-sectional observational study.

## 2. Materials and Methods

### 2.1. Study Design

This is a hospital-based cross-sectional study, where pediatric dialysis (HD and PD) patients or their parents/caregivers were administered an electronic questionnaire at a single point in time to determine the barriers to medication adherence and their relationship with determinants of health. This design is more cost-effective and efficient and provides a sufficient descriptive snapshot of the study population [22] and insights about factors affecting medication adherence across patient groups.

### 2.2. Study Setting

Using the simple random technique, eligible participants (aged <18 years or their parents/caregivers) were identified and contacted via the hospital electronic database (eSIHI) medical records and dialysis unit nurses at King Khalid University Hospital (KKUH). A validated questionnaire, including a medication adherence scale [23] was distributed electronically to participants (parents/caregivers or pediatric patients, if literate) after they were assured of confidentiality and gave written consent. The study followed the 1964 Helsinki Declaration guidelines and was approved by the King Saud University College of Medicine Institutional Review Board (Ethics Approval Number: E-22-7079).

### 2.3. Target Population and Sample Size

The target population was parents/caregivers of and/or children undergoing HD/PD who were exposed to drugs (medications) during dialysis. There were 78 pediatric patients undergoing HD/PD. The estimated sample size was based on a mean score of medication adherence of 30 ± 3 [18]. Therefore, assuming an expected population standard deviation of 3 with a 5% margin of error at 95% confidence level, the minimum required sample size was 70 to ensure adequacy, given the relative rarity of pediatric patients undergoing HD/PD [19].

### 2.4. Inclusion Criteria

Eligible participants were children (<18 years) undergoing HD/PD in the pediatric center at KKUH who could read and write Arabic. Questionnaires were self-administered by pediatric patients (if literate) but were also completed by parents/caregivers of younger illiterate children.

### 2.5. Study Variables

The main exposure variable was patients on HD or PD taking standard prescribed medications during HD/PD for anemia (iron, erythropoietin, darbepoetin, and folic acid), bones and minerals (calcium carbonate, calcitriol, alfacalcidol, vitamin D-cholecalciferol, sevelamer, and cinacalcet), and blood pressure (amlodipine, clonidine, lisinopril, prazosin, and sodium bicarbonate).

The primary outcome was adherence to medications, and factors (obstacles and barriers) limiting the use of medicines were measured. Sociodemographic variables (age, gender, residential area, education levels, family income, etc.) and common confounders were assessed.

### 2.6. Data Collection

The duration of the study was three months. Participants were emailed the questionnaires and given three weeks to complete them, plus a weekly reminder for an improved response rate. The questionnaire utilized a validated adherence scale [23] translated into Arabic by an expert and reviewed by a 12-member panel (nephrologists, epidemiologists, and consultants) against the English version, and suggestions were incorporated. A 10-participant pilot study at KKUH tested its clarity and understanding before finalizing the Arabic version.

The survey had three sections: patient demographics (age, sex, modality of care, education level, and medications used), parent/caregiver information (relationship, education, work status, and income), and a medication adherence score using a validated scale [23].

The 17-item Adolescent Medication Barriers Scale [23] was used to determine medication adherence barriers. The questionnaire had five statements about medication perceptions using a Likert scale. Values were coded as 0 = Strongly Disagree, 1 = Disagree, 2 = Not Sure/Sometimes, 3 = Agree, 4 = Strongly Agree. The 17-question adherence scale had total scores ranging from 0 to 68. Lower overall scores indicated fewer medication adherence barriers, while higher scores meant more barriers.

### 2.7. Data Management and Analysis

Data were cleaned and incomplete entries were removed. SPSS Statistics v26 (SPSS, Chicago, IL, USA) was used for statistical analysis. Categorical variables (gender, marital status) were expressed as counts or percentages. Numerical variables (age, number of children, adherence scores) were summarized as means and standard deviations (SD). Multivariable-adjusted regression was used to predict medication adherence adjusting for common confounders. The statistical significance and precision of the data were reported using a *p*-value of <0.05 and a 95% confidence interval (CI).

### 2.8. Ethics Approval 

The study followed the 1964 Helsinki Declaration guidelines and was approved by the King Saud University College of Medicine Institutional Review Board (Ethics Approval Number: E-22-7079). The participants gave written consent for participation and the publication of the study. 

## 3. Results

### 3.1. Sociodemographic and Clinical Results

Almost all the pediatric patients at KKUH (or their caregivers) participated in the study. Table 1 summarizes the sociodemographic characteristics of the respondents. Most respondents (85.9%) were parents/caregivers of the pediatric patients and 14.1% were the pediatric patients themselves. The mean age of the caregivers was 41.21 (±7.76) years, 39.7% of the caregivers had bachelor’s degrees, 66% were married, 52.6% were employed, and overall, the responses indicated lower-middle-class socioeconomic status. Most parents/caregivers (70.5%) did not have private insurance. There were <5 members in 52.6% families and a median of 3 children. Respondents lived outside (42.3%) and north of Riyadh city (23.1%), while 21.8% of respondents moved households because of their child’s kidney disease.

Table 2 summarizes the sociodemographic characteristics of children with kidney disease. The mean age of the children was 8.15 ± 4.96 years, 71.8% children were undergoing PD, and 56.4% children were uneducated either because they were underage for school or affected by their health condition (28.2% and 16.7%, respectively). The most and least reported concomitant morbidities were hereditary diseases and rheumatic diseases (43.6% and 3.8%, respectively).

Table 3 compares the medicines used by HD and PD pediatric patients. More patients on PD used medications (iron supplements, erythropoeitin, garbopoetin, folic acid, calcium carbonate, calcitriol, imlodobin, clonidine, alfacalcidolin, sodium bicarbonate, and cholecalciferol) compared to patients on HD. Only erythropoietin use showed statistically significant differences between patients on different dialysis modalities (*p*-value < 0.001).

### 3.2. Adherence Scores by Dialysis Modality

Table 4 shows a trend of higher medication adherence for PD patients compared with HD patients (36.1 ± 12.9 vs. 34.7 ± 8.3, *p* = 0.07). The leading medication adherence barriers for all patients were ‘tired of taking the medication’ (score = 3.0256), ‘difficulty swallowing pills’ (score = 2.8974), ‘did not feeling like taking medication sometimes’ (score = 2.7436), ‘tired of the medical condition’ (score = 2.5128), taste (score = 2.5513), and forgetfulness (score = 2.41). The most reported barriers among the PD patients compared to the HD patients were ‘not feeling like taking medication sometimes’ (score = 2.9643) and ‘too many pills’ (score = 3.21430).

Table 5 shows the results of univariate linear regression to assess factors that significantly affect medication adherence in the study population. The score significantly increased with an increase in age (*p* = 0.019) but significantly decreased if parents were employed (*p* = 0.037). Other confounding factors included the presence of type 1 diabetes, systematic lupus erythematous (SLE)/rheumatoid disease, widowed parents, and residing outside Riyadh. These factors were adjusted in the multiple linear regressions (multivariate) (Table 6). After adjusting for confounders, the adherence scores significantly increased with increasing age (β = 2.378, *p* < 0.001), employed parents (β = 8.726, *p* = 0.011), and living outside Riyadh (β = 19.198, *p* < 0.001).

## 4. Discussion

To the best of our knowledge, this is one of the few studies that has assessed medication adherence barriers and health determinants in pediatric patients undergoing HD/PD. There was a trend of higher medication adherence for children on PD (more barriers) than those on HD (36.1 ± 12.9 vs. 34.7 ± 8.3). The leading overall barrier was ‘tired of taking medications’ (score = 3.0256). Health determinants significantly affected adherence (*p*-value < 0.05). Children (56.4%) were uneducated (56.4%) about CKD and treatments like HD/PD (16.7%). After adjusting for common confounders, the medication adherence scores significantly increased (more barriers) with increased patient age (β = 2.378, *p* < 0.001), working parents (β = 8.726, *p* = 0.011), and those living outside Riyadh city (β = 19.198, *p* < 0.001).

### 4.1. Barriers to Medication Adherence

Medication adherence is an essential factor that influences pediatric HD/PD patients’ outcomes [5,24,25]. Our findings align with the literature indicating that forgetfulness, medication burdens, disease burdens, and unhealthy behaviors and attitudes reduce adherence and quality of life [10,26,27].

There were more adherence barriers for PD patients than HD patients, perhaps because HD offers some relief through treatments unavailable to PD patients [24]. Key PD barriers were ‘not feeling like taking medications sometimes’ (score = 2.9643) and too many pills (score = 3.21430). For HD patients, the most reported barriers were ‘tired of taking medicine’ (score = 3.0893) and too many pills (score = 2.4545). The number of pills and attractiveness of packaging also influenced adherence [16]. The extent of adverse effects (nausea, vomiting, and constipation) can increase non-adherence and non-attendance as patients weigh the benefits of medical adherence and adverse effects [5,24,28,29]. Experiencing multiple HD/PD sessions heightened fear and non-adherence. Therefore, healthcare providers must gain more insights about non-adherence [26,30] or adjust medications to promote adherence. Identifying and addressing barriers is key in improving adherence outcomes [31]. Examining the alignment of behavior and medical advice is complex in chronically ill children. Although up to 60% of people with kidney failure have poor adherence, there is lack of research about this [19,21].

### 4.2. Health Determinants Contributing to Medication Adherence

Patient-related, socioeconomic, therapy-related, and condition-related factors are globally linked to medication non-adherence [28]. We found that adherence increased with age, aligning with evidence of higher adherence in older patients undergoing HD/PD, which can be explained by the poor understanding of the severity of the condition [22,28]. Severity and disability levels also affect adherence [28].

Socioeconomic status influences medication adherence [28,32]. Patients in our study had lower-middle-class socioeconomic status (income SAR 5000–10,000 per month) (38.5%), and lived outside Riyadh city (42.3%) (not statistically significant but negatively associated). Children of working parents had more adherence barriers (β = 8.726, *p* = 0.011). Social disparities limit healthcare access and expenses, affecting adherence and wellbeing [33,34]. Many patients in our study had lower-middle-class socioeconomic status, limited access to healthcare interventions to control the adverse effects of chronic diseases, treatment costs, and 70.5% of respondents had no private health insurance.

Medication non-adherence rates affect outcomes, costs [35,36,37], risk treatment failure, infectious disease, and mortality [38], reduce clinical understanding [39], and can lead to comorbidities and lower survival rates [25]. If patients demonstrate medication non-adherence, physicians may avoid changing regimens and recommending transplants with strict post-operative medication requirements [39]. More research is needed about healthcare provider-related factors, as complex prescriptions and inadequate patient education on risks/benefits contributes to poor adherence [4,32,40]. In this study, 16.7% of children were uneducated and 21.8% kept moving houses to accommodate their condition and its frequent long-term treatment. There is evidence indicating that increased home mobility throughout the early years of life can have a negative impact on a child’s mental health (in later life) [41] and education [42].

Treatment duration, complexity, and adverse effects influence adherence. Long HD/PD routines cause boredom and the desire to quit. Hypotension, cramps, and pain increase non-adherence [29]. Recognizing the negative effects of non-adherence and addressing identified barriers and patients’ medication beliefs are important to improve adherence and patient outcomes.

The choice between home PD and HD for patients with kidney failure significantly impacts medication adherence. Patients undergoing home PD often experience greater autonomy and flexibility in managing their treatment regimen compared to those undergoing in-center HD. This increased control over their dialysis schedule and medication administration may lead to improved adherence to prescribed medications [43]. However, home PD is underutilized in Saudi Arabia [44]. Conversely, patients undergoing in-center HD may face logistical challenges, such as transportation to and from dialysis centers and adherence to fixed treatment schedules, which could potentially hinder medication adherence. Furthermore, the complexity of the dialysis procedure itself, along with associated symptoms such as fatigue and discomfort, may also influence medication adherence among HD patients. Thus, understanding the unique factors influencing medication adherence in patients undergoing home PD versus in-center HD is essential for optimizing treatment outcomes and enhancing the quality of care for individuals with kidney failure.

The inadequate development of arteriovenous fistula in pediatric patients, resulting from an underdeveloped arterial vasculature and challenges in angiogenesis [45], combined with healthcare access barriers, contributes to the relative scarcity of pediatric patients undergoing HD as compared to adult counterparts. Effectively addressing the intricacies inherent in pediatric patients undergoing HD/PD requires a comprehensive comprehension of the obstacles imposed by healthcare barriers. The constrained availability of specialized pediatric dialysis facilities, exacerbated by socioeconomic constraints, further diminishes the frequency of pediatric HD/PD utilization in contrast to the adult demographic.

The findings from the research on low medical adherence among pediatric patients undergoing HD/PD and the factors influencing adherence, such as family dynamics, physiological aspects, and comorbidities, hold significant implications for healthcare practice and policy. Particularly, in developing nations, where economic burdens, limited access to medications, and healthcare challenges compound the issue, understanding these barriers becomes crucial. The existing scholarly literature accentuates the imperative for pioneering interventions designed to optimize medication adherence among pediatric patients undergoing HD/PD [28,32,35]. Subsequent initiatives should prioritize the formulation of sophisticated systematic frameworks and the implementation of high-caliber digital tools, capitalizing on the dynamic landscape of technology. Embracing these progressive strategies facilitates the medical community’s navigation toward augmented outcomes and an enhanced quality of life for pediatric patients undergoing HD/PD.

### 4.3. Strengths and Limitations

KKUH, Saudi Arabia offers specialized CKD management for pediatric patients and despite healthcare barriers affecting access, the hospital provides social support, clinical care, and medications to CKD patients. Therefore, the study sample had a good cross-section of participants.

Limitations include the cross-sectional design and small sample size, as pediatric HD/PD is less common than adult dialysis [19]. Cross-sectional studies, while valuable for capturing a snapshot of a population at a specific point in time, come with inherent limitations that can impact the study’s generalizability. One significant limitation is the reliance on self-reported data, which introduces the potential for recall bias. Participants may not accurately remember or may provide socially acceptable responses, compromising the accuracy of the information collected. Although medication adherence was assessed at a single time point allowing a snapshot relationship assessment between outcomes and medication adherence, this approach might minimize the chance of recall bias associated with medication adherence assessment for a longer duration. Indeed, medication adherence benefits among children were reported in clinical trials that assessed adherence over a short period of time (i.e., three months) [46]. Another limitation is that data pertaining to the administration route of erythropoietin was not collected in our study. However, our study provides robust insights into medication adherence among pediatric patients undergoing HD/PD.

## 5. Conclusions

This study found some barriers for patients undergoing PD and HD, indicating shared obstacles across different dialysis modalities and other chronic diseases. These findings can be integrated with existing practices and inform long-term initiatives to decrease medication non-adherence and ensure targeted, evidence-based interventions. Support strategies and patients’ education should consider whether non-adherence is intentional or unintentional. For example, forgetfulness may require behavioral interventions like reminder alarms. As patients rarely voice non-adherence, professionals should be encouraged to raise issues with hospital management and the government to create an environment to discuss these challenges. Digital health technologies offer cost-effective supplementary strategies for managing adherence. Future efforts to improve pediatric patient medication adherence should focus on leveraging systematic frameworks and quality digital tools, given the increasing availability of technology. Incorporating innovative strategies such as mobile applications, interactive platforms, and personalized interventions can enhance engagement and adherence among pediatric patients. Additionally, the implementation of randomized trials addressing medication adherence among children will provide valuable insights into the efficacy of specific interventions, guiding evidence-based approaches for optimizing adherence in this population. This proactive approach aligns with the evolving landscape of healthcare technology and underscores the importance of evidence-driven strategies in promoting successful medication adherence among pediatric patients.

## Figures and Tables

**Table 1 children-11-00293-t001:** Sociodemographic variables of the respondents (N = 78).

Variables		N = 78
Relationship to dialysis patient	I’m the one who does dialysis	11 (14.1%)
	Mother	42 (53.8%)
	Child’s caregiver	13 (16.7%)
	Father	12 (15.4%)
Age of the caregiver (years)	Mean ± SD (Min–Max) years	41.21 ± 7.76 (20–60)
Education level of the parent/caregiver	Illiterate	7 (9%)
	Primary school	1 (1.3%)
	Middle school	2 (2.6%)
	High school	18 (23.1%)
	Diploma	14 (17.9%)
	Bachelor’s degree	31 (39.7%)
	Master’s or PhD	5 (6.4%)
Caregiver’s marital status	Married	66 (84.6%)
	Widowed	5 (6.4%)
	Divorced or separated	7 (9%)
Caregiver’s work status	Not working	37 (47.4%)
	Working	41 (52.6%)
Monthly family income (Saudi riyal)	<5000 Saudi riyals	20 (25.6%)
	5000–10,000 Saudi riyals	30 (38.5%)
	>10,000 Saudi riyals	28 (35.9%)
Private insurance	No	55 (70.5%)
	Yes	23 (29.5%)
No. of family members	<5	41 (52.6%)
	>5	37 (47.4%)
No. of children	Median (min–max)	3 (1–8)
Residency region	Northern Riyadh	18 (23.1%)
	Southern Riyadh	9 (11.5%)
	Western Riyadh	9 (11.5%)
	Eastern Riyadh	9 (11.5%)
	Outside Riyadh city	33 (42.3%)
Moved house due to kidney failure	No	61 (78.2%)
	Yes	17 (21.8%)

Max, maximum; Min, minimum; SD, standard deviation.

**Table 2 children-11-00293-t002:** Sociodemographic variables of the dialysis patients (N = 78).

Variables		N = 78
Age (child) in years	Mean ± SD	8.15 ± 4.96
Sex of the child	Male	39 (50.0%)
	Female	39 (50.0%)
Modality of dialysis	Hemodialysis	22 (28.2%)
	Peritoneal dialysis	56 (71.8%)
Child’s education level	No education	44 (56.4%)
	Primary school	18 (23.1%)
	Middle school	11 (14.1%)
	High school	5 (6.4%)
Reason for being uneducated (N = 44)	Underage for school	22 (28.2%)
	Health condition	13 (16.7%)
	Not reported	9 (11.5%)
Chronic diseases (>1 disease per child)	Type 1 Diabetes	14 (17.9%)
	Genetic diseases	24 (30.8%)
	Hereditary diseases	34 (43.6%)
	Hypertension	25 (32.1%)
	Nephrotic syndrome	22 (28.2%)
	SLE/rheumatic diseases	3 (3.8%)

SD, standard deviation; SLE, systemic lupus erythematosus.

**Table 3 children-11-00293-t003:** Medications used for the dialysis patients (N = 78).

Variables		Medications Used with Dialysis			
		Total	Hemodialysis N = 22 (28.2%)	Peritoneal Dialysis N = 56 (71.8%)	*p*-Value
Iron	No	21 (26.9%)	4 (18.2%)	17 (30.4%)	0.27
	Yes	57 (73.1%)	18 (81.8%)	39 (69.6%)	
Erythropoietin	No	18 (23.1%)	11 (50.0%)	7 (12.5%)	0.001 *
	Yes	60 (76.9%)	11 (50.0%)	49 (87.5%)	
Darbepoetin	No	28 (35.9%)	9 (40.9%)	19 (33.9%)	0.56
	Yes	50 (64.1%)	13 (59.1%)	37 (66.1%)	
Folic acid	No	26 (33.3%)	7 (31.8%)	19 (33.9%)	0.86
	Yes	52 (66.7%)	15 (68.2%)	37 (66.1%)	
Calcium carbonate	No	20 (25.6%)	5 (22.7%)	15 (26.8%)	0.71
	Yes	58 (74.4%)	17 (77.3%)	41 (73.2%)	
Calcitriol	No	34 (43.6%)	13 (59.1%)	21 (37.5%)	0.08
	Yes	44 (56.4%)	9 (40.9%)	35 (62.5%)	
Alfacalcidol	No	18 (23.1%)	5 (22.7%)	13 (23.2%)	0.96
	Yes	60 (76.9%)	17 (77.3%)	43 (76.8%)	
Cholecalciferol	No	17 (21.8%)	5 (22.7%)	12 (21.4%)	0.90
	Yes	61 (78.2%)	17 (77.3%)	44 (78.6%)	
Sevelamer	No	28 (35.9%)	7 (31.8%)	21 (37.5%)	0.64
	Yes	50 (64.1%)	15 (68.2%)	35 (62.5%)	
Cinacalcet	No	46 (59.0%)	13 (59.1%)	33 (58.9%)	0.99
	Yes	32 (41.0%)	9 (40.9%)	23 (41.1%)	
Imlodobin	No	23 (29.5%)	9 (40.9%)	14 (25.0%)	0.16
	Yes	55 (70.5%)	13 (59.1%)	42 (75.0%)	
Clonidine	No	39 (50.0%)	14 (63.6%)	25 (44.6%)	0.13
	Yes	39 (50.0%)	8 (36.4%)	31 (55.4%)	
Lisinopril	No	43 (55.1%)	9 (40.9%)	34 (60.7%)	0.11
	Yes	35 (44.9%)	13 (59.1%)	22 (39.3%)	
Prazocin	No	51 (65.4%)	16 (72.7%)	35 (62.5%)	0.39
	Yes	27 (34.6%)	6 (27.3%)	21 (37.5%)	
Sodium bicarbonate	No	30 (38.5%)	12 (54.5%)	18 (32.1%)	0.06
	Yes	48 (61.5%)	10 (45.5%)	38 (67.9%)	

* *p*-value < 0.05.

**Table 4 children-11-00293-t004:** Barriers to medication adherence scores among dialysis patients.

Statement	Peritoneal Dialysis (N = 56)	Hemodialysis (N = 22)	ALL (N = 78)
I don’t want to take the medicine at school	1.0000	1.0455	1.0128
I feel that it gets in the way of my activities.	1.5357	1.4545	1.5128
I am forgetful and I don’t remember to take the medicine every time.	2.3214	2.6364	2.4103
I do not want other people to notice me taking the medicine.	1.8214	1.7273	1.7949
I sometimes just don’t feel like taking the medicine.	2.9643	2.1818	2.7436
I don’t like what the medication does to my appearance.	1.9286	1.7273	1.8718
I am tired of taking medicine.	3.0893	2.8636	3.0256
I am tired of living with a medical condition.	2.5000	2.5455	2.5128
I believe the medicine is hard to swallow.	2.9643	2.7273	2.8974
I believe that I have too many pills to take.	3.2143	2.5909	3.0385
I don’t like how the medicine tastes.	2.5893	2.4545	2.5513
I believe the medicine has too many side-effects.	1.8750	1.8636	1.8718
I get confused about how the medicine should be taken (with or without food, with or without water etc.).	1.8571	2.5000	2.0385
I am not organized about when and how to take the medicine.	2.3393	2.3636	2.3462
I find it hard to stick to a fixed medication schedule.	2.3214	2.3636	2.3333
Sometimes I don’t realize when I run out of pills	1.5179	1.6364	1.5513
Sometimes it’s hard to make it to the pharmacy to pick up the prescription before the medicine runs out.	1.0179	2.0000	1.2949
Total Score * Mean adherence score	2.241 36.1 ± 12.9	2.158 34.7 ± 8.3	2.165 36.70 ± 12.97

* *p* = 0.07 for the mean adherence score for peritoneal dialysis and hemodialysis.

**Table 5 children-11-00293-t005:** Univariate linear regression for factors affecting adherence scores.

Independent Variables	Reference Category	β	*p*-Value	95% CI for β
Age of the child	Numerical	0.704	0.019 *	0.120 to 1.289
Female sex of the child	Male	−1.051	0.723	−6.936 to 4.834
Peritoneal dialysis	Hemodialysis	0.286	0.931	−6.258 to 6.830
Highest school grade child completed	Ordinal	1.704	0.281	−1.421 to 4.828
Child has Type 1 diabetes	No	7.150	0.061	−0.348 to 14.648
Child has genetic diseases	No	−1.801	0.575	−8.168 to 4.566
Child has hereditary diseases	No	4.068	0.171	−1.797 to 9.934
Child has hypertension	No	−1.214	0.702	−7.519 to 5.090
Child has nephrotic syndrome	No	−4.211	0.199	−10.684 to 2.262
Child has SLE/rheumatic diseases	No	−13.560	0.076	−28.557 to 1.437
Iron medication prescribed	No	1.877	0.574	−4.748 to 8.503
Age of caregiver/parent	Numerical	0.192	0.316	−0.187 to 0.571
Divorced/separated parents	Married or widowed	1.893	0.715	−8.401 to 12.188
Widowed parent	Married or divorced	9.932	0.098	−1.876 to 21.739
Residence outside Riyadh	In Riyadh	5.081	0.088	−0.766 to 10.927
Parents’ education level	Ordinal	−0.293	0.754	−2.149 to 1.562
Working parent/caregiver	Not working	−6.114	0.037 *	−11.844 to −0.384
Family’s monthly income (range)	Ordinal	−0.628	0.742	−4.412 to 3.156
Had private insurance	No	−0.137	0.966	−6.595 to 6.321
Family members ≥ 5	<5	−0.313	0.916	−6.210 to 5.584
Number of children	Numerical	0.625	0.437	−0.967 to 2.216
Moved households due to kidney failure	No	1.054	0.769	−6.075 to 8.183

β, regression coefficient; CI, confidence interval for the regression coefficient; SLE, systemic lupus erythematosus; Type 1 diabetes * *p* < 0.05.

**Table 6 children-11-00293-t006:** Multiple linear regression for factors affecting the adherence score.

Independent Variables	Reference Category	β	*p*-Value	95% CI for β
Age of the child	Numerical	2.378	<0.001 *	1.773 to 2.983
Child has Type 1 DM	No	2.296	0.657	−7.963 to 12.556
Child has SLE/rheumatic diseases	No	10.744	0.280	−8.952 to 30.440
Working parent/caregiver of the child	Not working	8.726	0.011 *	2.099 to 15.353
Widowed parent	Married/divorced	7.638	0.355	−8.708 to 23.983
Residence outside Riyadh	In Riyadh	19.198	<0.001 *	12.493 to 25.903

β, regression coefficient; CI, confidence interval for the regression coefficient; SLE, systemic lupus erythematosus; Type 1 diabetes; * *p* < 0.05.

## Data Availability

The data presented in this study are available on request from the corresponding author. The data are not publicly available due to patients’ privacy and the Institutional Review Board’s rules and regulations.

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
