# Peer review of "Medication Adherence Barriers and Their Relationship to Health Determinants for Saudi Pediatric Dialysis Patients"

_children, 2024, doi:10.3390/children11030293_

Round 1

Reviewer 1 Report

Comments and Suggestions for Authors

Overall, the manuscript is well-crafted and addresses a significant gap in the existing literature. However, before considering it for publication, several significant omissions should be addressed.

Introduction:

While the introduction provides a thorough overview of the importance of medication adherence in chronic diseases, it would be beneficial to clearly state the specific gap or need that this study aims to address.

Three sentences with similar content exist in lines 57-58, 63, and 66-67, requiring attention to enhance clarity and avoid repetition.

Methods:

In lines 95-96, it is unclear what "30±3" refers to in the sentence "The estimated sample size was based on a mean medication adherence of 30±3." Please provide clarification.

The calculated sample size was 70 patients, yet 78 were included in the study. Please explain this discrepancy.

More information on the study's duration and the specific time frame for data collection would enhance the manuscript.

Clarify whether the questionnaire was self-administered by pediatric patients (if literate) or completed by their parents/caregivers.

Specify the types of medications involved when mentioning exposure to drugs during dialysis in the inclusion criteria.

Provide additional details on the statistical methods used, particularly regarding the adjustment of common confounders in multiple linear regressions.

Discussion:

Include a discussion on the clinical implications of the findings and potential interventions based on the identified barriers.

While the study's strengths are mentioned, they are somewhat uninformative. Address other limitations, such as the reliance on self-reported data and potential sources of bias, explaining their impact on the study's generalizability.

Offer recommendations for future research directions in this field, taking into account the limitations of the current study.

Reference List:

More than 40% of the cited sources are older than 10 years. To ensure the inclusion of the latest research findings, consider adding recent references (after 2022).

Finally, revise the text to describe patients without explicitly referring to the treatment options. For instance, use the term "patients undergoing HD/PD" instead of the term "dialysis patients," in alignment with KDIGO nomenclature.

Comments on the Quality of English Language

The overall English quality of the paper is quite good. However, minor English polishing is required.

Author Response

Reviewer 1

A. Introduction:

  1. While the introduction provides a thorough overview of the importance of medication adherence in chronic diseases, it would be beneficial to clearly state the specific gap or need that this study aims to address.

Thank you for your comment. Paragraphs about the gap in literature and the aim of the study were rewritten on page no.2, lines 73- 89.

  1. Three sentences with similar content exist in lines 57-58, 63, and 66-67, requiring attention to enhance clarity and avoid repetition.

Thank you for your comment. The paragraphs about the gap in literature and the aim of the study were rewritten on page no.2, lines 73- 89.  

B. Methods:

  1. In lines 95-96, it is unclear what "30±3" refers to in the sentence "The estimated sample size was based on a mean medication adherence of 30±3." Please provide clarification.

Thank you for your comment. Yes, the sample size calculation was based on the mean score of medication adherence (30±3) reported by previous study [18]. The sentence was revised as requested (page no.3, lines 110-114).

  1. The calculated sample size was 70 patients, yet 78 were included in the study. Please explain this discrepancy.

Thank you for your inquiry regarding the deviation from the initially calculated sample size. While our initial calculation suggested a sample size of 70 patients, we ultimately opted to include 78 participants in the study. As the calculation for sample size provides an approximation of the minimum required sample, this decision was made to enhance the robustness and generalizability of our findings (https://doi.org/10.4103%2F0301-4738.71673 ).

By surpassing the calculated sample size, we aimed to account for potential unforeseen variations and ensure a more comprehensive representation of the target population. This strategic approach allows for a more reliable statistical analysis, increased power, and a greater ability to detect subtle effects.

We believe that the inclusion of extra participants will contribute to the overall strength of our study, providing a more nuanced and accurate understanding of the investigated variables.

  1. More information on the study's duration and the specific time frame for data collection would enhance the manuscript.

Thank you for your comment. The study duration (i.e. data collection time) was about 3 months; this information is added to the methodology (page no.3, line 121). The time frame for data collection was given in the data collection section (page no.3, lines 121- 123): “participants were given three weeks to complete questionnaire, plus a weekly reminder…”

  1. Clarify whether the questionnaire was self-administered by pediatric patients (if literate) or completed by their parents/caregivers.

Thank you for the comment. Questionnaires were self-administered by pediatric patients (if literate) but were also completed by parents/caregivers of younger illiterate children. This information was updated on the methodology page no. 3, lines 117-119.

  1. Specify the types of medications involved when mentioning exposure to drugs during dialysis in the inclusion criteria.

Thank you for your comment. Details about medications used during dialysis are available under section 2.4. Study variables: “The main exposure variable was patients on HD or PD taking standard prescribed medications during dialysis for anemia (iron, erythropoietin, darbepoetin, and folic acid), bones and minerals (calcium carbonate, calcitriol, alfacalcidol, vitamin D-cholecalciferol, sevelamer, and cinacalcet), and blood pressure (amlodipine, clonidine, lisinopril, prazosin, and sodium bicarbonate)” on page no. 3, lines 121-125.

  1. Provide additional details on the statistical methods used, particularly regarding the adjustment of common confounders in multiple linear regressions.

Thank you for your comment. The analysis section was updated with details about regression analysis as requested on page no.4, lines 152-155.

C. Discussion:

  1. Include a discussion on the clinical implications of the findings and potential interventions based on the identified barriers.

Thank you for the suggestions. The discussion was updated with a paragraph about implications based on the finding of this study on pages no.9-10, lines 297- 308.

  1. While the study's strengths are mentioned, they are somewhat uninformative. Address other limitations, such as the reliance on self-reported data and potential sources of bias, explaining their impact on the study's generalizability.

Thank you for your comment. The limitations section was updated as requested on page no. 9-10, lines 291- 302.

  1. Offer recommendations for future research directions in this field, taking into account the limitations of the current study.

Thank you for your suggestion. A paragraph was added to the recommendation section on page no.10, lines 315- 329.

D. Reference List:

  1. More than 40% of the cited sources are older than 10 years. To ensure the inclusion of the latest research findings, consider adding recent references (after 2022).

Thank you for your insightful comments. We have incorporated new references as suggested. However, it remains crucial to include studies published before 2022, as there is a scarcity of evidence pertaining to medication adherence in the pediatric population undergoing HD/PD. We are committed to addressing this gap by integrating relevant earlier studies to ensure a comprehensive and well-grounded literature review. Your guidance is invaluable in refining the robustness of our research.

E. Finally, revise the text to describe patients without explicitly referring to the treatment options. For instance, use the term “patients undergoing HD/PD” instead of the term “dialysis patients,” in alignment with KDIGO nomenclature.

Thank you for your comment. The manuscript was updated with “patients undergoing HD/PD” as suggested.

F. The overall English quality of the paper is quite good. However, minor English polishing is required.

Thank you for your comment. The manuscript has been professionally edited (please see attached editing certificate).

Reviewer 2 Report

Comments and Suggestions for Authors

This study examines medication adherence barriers and their relationship to health determinants among Saudi Arabia children on dialysis, to enhance treatment success.

The language used throughout the text seems correct and the references are in general relevant, appropriate and up-to-date.

Indeed, medication adherence is critical for treatment and improved outcomes of chronic diseases. Indeed, evidence-based targeted interventions can increase medication adherence in this group on frequent dialysis. 

However, this hospital-based cross-sectional observational study presents several limitations, as mentioned, mainly due to its nature design as the small size and as the medication adherence was assessed at a single time point allowing only a snapshot relationship assessment between outcomes and medication adherence. Moreover, it must be taken into account a lack of maturation of arteriovenous fistula due to an immature vasculature arterial system and difficulties in angiogenesis for pediatric patients and healthcare barriers affecting access, making pediatric dialysis less common than adult dialysis. I would personally like that authors add 3-4 lines-sentences before "Conclusions" concerning this last problem in pediatric dialysis and their own opinion based on current existing literature focusing on possible future efforts to improve pediatric patient medication adherence, leveraging systematic frameworks and quality digital tools, given the increasing availability of technology in order to better describe future perspectives of this interesting clinical model.

Author Response

Reviewer 2

Comments and Suggestions for Authors

  1. This study examines medication adherence barriers and their relationship to health determinants among Saudi Arabia children on dialysis, to enhance treatment success. The language used throughout the text seems correct and the references are in general relevant, appropriate and up-to-date. Indeed, medication adherence is critical for treatment and improved outcomes of chronic diseases. Indeed, evidence-based targeted interventions can increase medication adherence in this group on frequent dialysis. 

Thank you for your gracious remarks; they are greatly appreciated.

  1. However, this hospital-based cross-sectional observational study presents several limitations, as mentioned, mainly due to its nature design as the small size and as the medication adherence was assessed at a single time point allowing only a snapshot relationship assessment between outcomes and medication adherence.

We appreciate your thoughtful comment and acknowledge the inherent limitation of cross-sectional studies in capturing exposure and outcomes at a singular time point. Nevertheless, it is noteworthy to mention that in clinical trials evaluating medication adherence among children, positive benefits have been reported even when adherence was assessed over a relatively short duration [46].

To address concerns regarding recall bias associated with prolonged medication adherence assessment, we have incorporated the pertinent details into the limitations section on page 10, specifically on lines 315- 329. By employing a self-report questionnaire for medication adherence assessed at a single point in time, we aim to mitigate the potential for recall bias while recognizing the constraints of the cross-sectional study design.

  1. Moreover, it must be taken into account a lack of maturation of arteriovenous fistula due to an immature vasculature arterial system and difficulties in angiogenesis for pediatric patients and healthcare barriers affecting access, making pediatric dialysis less common than adult dialysis.

I would personally like that authors add 3-4 lines-sentences before "Conclusions" concerning this last problem in pediatric dialysis and their own opinion based on current existing literature focusing on possible future efforts to improve pediatric patient medication adherence, leveraging systematic frameworks and quality digital tools, given the increasing availability of technology in order to better describe future perspectives of this interesting clinical model.

Thank you for your suggestions. The comments were considered, and the manuscript was updated with the following paragraphs on page no.9 lines 288-296, and on pages no.9- 10, lines 297- 308.

The inadequate development of arteriovenous fistula in pediatric patients, resulting from an underdeveloped arterial vasculature and challenges in angiogenesis [45], combined with healthcare access barriers, contributes to the relative scarcity of pediatric patients undergoing HD as compared to its adult counterpart. Effectively addressing the intricacies inherent in pediatric patients undergoing HD/PD requires a comprehensive comprehension of the obstacles imposed by healthcare barriers. The constrained availability of specialized pediatric dialysis facilities, exacerbated by socio-economic constraints, further diminishes the frequency of pediatric HD/PD utilization in contrast to the adult demographic.

The findings from the research on low medical adherence among pediatric patients undergoing HD/PD and the factors influencing adherence, such as family dynamics, physiological aspects, and comorbidities, hold significant implications for healthcare practice and policy. Particularly, in developing nations, where economic burdens, limited access to medications, and healthcare challenges compound the issue, understanding these barriers becomes crucial. Existing scholarly literature accentuates the imperative for pioneering interventions designed to optimize medication adherence among pediatric patients undergoing HD/PD [28, 32, 35]. Subsequent initiatives should prioritize the formulation of sophisticated systematic frameworks and the implementation of high-caliber digital tools, capitalizing on the dynamic landscape of technology. Embracing these progressive strategies facilitates the medical community's navigation toward augmented outcomes and an enhanced quality of life for pediatric patients undergoing HD/PD.

Reviewer 3 Report

Comments and Suggestions for Authors

Dear colleagues. Thank you for the opportunity to review your interesting paper. I found the paper original , and important idea for publication. However, there are some revisions I believe will improve the article.

  1. Introduction: Please add information about dialysis in Saudi Arabia and dialysis in children.
  2. In the introduction, it would be beneficial to also include information about the challenges faced by children and their parents undergoing dialysis.
  3. Methods: Did all participants know how to read and write? Were all children treated with dialysis included in this study?
  4. Over what period of time? Please elaborate on the questionnaire and specify if it is valid.
  5. Results: Present the characteristics of the patients included in the study in Table 1, including background diseases, dialysis vintage, and laboratory tests at the time of study entry.
  6. I recommend presenting at least some of the results as figures and not just as tables.
  7. Please address whether there was a difference in response to erythropoietin injection compared to orally taken medications.
  8. Children aged 3-5 depend on a caregiver, while children aged 12 and older are adolescents and should be treated differently, with results analyzed separately. Educational years are also important to note and analyze.
  9. The discussion is very sparse and it's important to also refer to the existing literature and analyze all results. Additionally, there is an inherent difference between patients undergoing home peritoneal dialysis and those opting for hemodialysis, which also needs to be discussed.
  10. Please add P values to the abstract.
Comments on the Quality of English Language

Moderate editing of English language required

Author Response

Reviewer 3

Comments and Suggestions for Authors

  1. Introduction: Please add information about dialysis in Saudi Arabia and dialysis in children.

Thank you for your comment. The introduction was updated with additional details about dialysis and children in Saudi Arabia on page no. 1- 2, lines 41- 50.

  1. In the introduction, it would be beneficial to also include information about the challenges faced by children and their parents undergoing dialysis.

Thank you for your comment. Challenges and barriers were added to the introduction on page no.2, lines 48-50 and lines 73- 79.

  1. Methods: Did all participants know how to read and write? Were all children treated with dialysis included in this study?

Thank you for your feedback. Indeed, all children undergoing treatment with either hemodialysis or peritoneal dialysis were included in our study. The questionnaires were designed to be self-administered by pediatric patients, if literate. For younger children unable to read or write, the parents or caregivers took on the responsibility of completing the questionnaires, as outlined in detail on page 3, lines 115-119 of the manuscript.

  1. Over what period of time? Please elaborate on the questionnaire and specify if it is valid.

T The study spanned a duration of three months, as explicitly stated on page 3, line 131. The questionnaire used in our research was validated and has been utilized in prior studies. For comprehensive information regarding the questionnaire and the associated adherence score, please refer to page 3, lines 133- 147.

  1. Results: Present the characteristics of the patients included in the study in Table 1, including background diseases, dialysis vintage, and laboratory tests at the time of study entry.

Thank you for your feedback. The characteristics of parents/caregivers are detailed in Table 1, outlined on pages 4 and 5, spanning lines 162 to 174. Additionally, Table 2 provides an overview of the characteristics specific to the children, as described on page 5, lines 175-178.

  1. I recommend presenting at least some of the results as figures and not just as tables.

Thank you for your suggestion to present some of the results as figures rather than exclusively in tables. While we appreciate the value of visual representation, we have opted for a comprehensive tabular format to ensure clarity and precision in conveying the study outcomes. This approach was chosen to maintain a concise and focused presentation of the data, facilitating a straightforward interpretation for the readers. However, we have ensured that the tables are appropriately labeled, and detailed descriptions are provided to enhance their accessibility. We believe this approach aligns with the nature of our data and contributes to the overall coherence of the manuscript. We appreciate your understanding of our decision and are open to further discussion on this matter.

  1. Please address whether there was a difference in response to erythropoietin injection compared to orally taken medications.

Thank you for your valuable feedback. Regrettably, specific data regarding the administration protocol of erythropoietin was not included in our study. We have duly recognized this limitation and acknowledged it in our manuscript on page (10), lines (326- 329). Despite this limitation, our study provides robust insights into medication adherence among pediatric patients undergoing HD/PD, and we appreciate the opportunity to enhance the clarity and comprehensiveness of our research in future endeavors.

  1. Children aged 3-5 depend on a caregiver, while children aged 12 and older are adolescents and should be treated differently, with results analyzed separately. Educational years are also important to note and analyze.

We appreciate your valuable feedback. Recognizing the significance of age and education as influential factors, it is crucial to note that the questionnaire was predominantly completed by parents or caregivers, with only 14% of children directly participating in the process. Importantly, our regression model carefully considered these factors, and confounding variables were effectively adjusted for in the analysis to mitigate their potential impact. Details regarding this analysis, including tables 5 and 6 and lines 201-215, can be found on page 7-8 of the manuscript.

  1. The discussion is very sparse and it's important to also refer to the existing literature and analyze all results. Additionally, there is an inherent difference between patients undergoing home peritoneal dialysis and those opting for hemodialysis, which also needs to be discussed.

Thank you for your valuable input. We have incorporated your suggestions into the discussion section, providing additional information and comparing HD and PD on page 9, lines 275-287. Furthermore, we have addressed the impact of inadequate development of arteriovenous fistula in pediatric patients on page 9, lines 288-296.

  1. Please add P values to the abstract.

Thank you for your comment. P- values were added as suggested.

  1. Comments on the Quality of English Language. Moderate editing of English language required

Thank you for your comment. The manuscript has been professionally edited (please see attached editing certificate).

Round 2

Reviewer 2 Report

Comments and Suggestions for Authors

Published in its revised form.

Author Response

(The authors gave the same response as above.)
